# Analysis of the Orchidaceae Diversity in the Pululahua Reserve, Ecuador: Opportunities and Constraints as Regards the Biodiversity Conservation of the Cloud Mountain Forest

**DOI:** 10.3390/plants11050698

**Published:** 2022-03-04

**Authors:** Mariana Mites, Herminia García-Mozo, Carmen Galán, Edwin Oña

**Affiliations:** 1International Campus of Excellence on Agreefood CeiA3, Andalusian Inter-University Institute for Earth System Research, University of Cordoba, 14071 Córdoba, Spain; z82micam@uco.es (M.M.); bv1gasoc@uco.es (C.G.); 2Ministerio del Ambiente, Quito 170525, Ecuador; edwin.ona@ambiente.gob.ec

**Keywords:** Orchidaceae, richness, diversity, cloud mountain forest

## Abstract

The Pululahua Geobotanical Reserve is a protected natural area in the cloud mountain forest of Ecuador, so rich in orchid species despite being a volcanic area still under threat of volcanic activity. A comparative biodiversity study of orchids was carried out in two different sectors, Chaupisacha (CH) and La Reventazón (LR). Data were collected in 1 ha plots in each sector, in which all the orchids found were counted and two individuals of each species were retained. Immature individuals were conserved in a plant nursery until flowering. In CH, there were 922 individuals grouped into 24 genera and 55 species, while LR had 9196 individuals grouped into 26 genera and 46 species; only 14 species were found in both sectors. Different density and diversity indexes were calculated. The density (ind./100 m^2^) of CH was 0.96, while that of LR was 185.92. Simpson’s diversity (1 − λ) attained CH 0.903 ± 0.01 and LR 0.85 ± 0.01. The orchid diversity measured by the Shannon-Wiener diversity index (H′) was 1.29 for CH, differing significantly from that of LR (H′ 1.02). The medium equity (Jaccard’s J′) found was 0.61 in CH and 90.78 in LR. Limitations as regards the natural dispersion of orchids seemed to favor endemism. Some species, such as *Dracula felix* and *Restrepia guttulate*, are threatened with disappearance from the wild or are vulnerable, as is the case for *Epidendrum polyanthogastrium*. A lack of information on the phenology and anthropic impacts in the area limit the conservation of species, signifying that new protected figures and seed banks are necessary, especially in CH, owing to its high diversity of orchids.

## 1. Introduction

The Pululahua Geobotanical Reserve in Ecuador has been identified as an area containing an important diversity of plants, despite being a small, protected area (3356 ha) within the PANE (Heritage of Natural Areas of the State) [1]. Endara [2] shows that Ecuador has a high diversity of species, with the western and eastern foothills of the Andes Mountain range (300–3000 m.a.s.l.) being the areas with the greatest richness in the country as regards the Orchidaceae family.

Despite being a small country, Ecuador has a high diversity of habitats and microclimates that favor biodiversity. This country is one of the 17 megadiverse countries in the world [3,4]. In fact, it has more biodiversity per square kilometer than any other country [5]. In the case of orchid diversity, several factors, such as the height of the Andes Mountain range, the influence of the warm and cold atmospheric ocean currents (El Niño and Humboldt, respectively), and a heavy rainfall regime, determine the high proliferation of these species in this country [6].

There is currently a register of 4187 species of orchids in Ecuador, of which 1707 are endemic [2,7]. According to herbaria reviews, most of the Ecuadorian orchid species live in a range of between 300 and 3000 m.a.s.l., while 588 species, i.e., 18% of the country’s orchid population, have been reported at higher elevations [8]. The areas in the country with the greatest richness as regards the Orchidaceae family are the western and eastern foothills of the Andes Mountain range.

It is important to state that orchids are an extensive plant group and there are perennial, epiphytic, and terrestrial herbs in different habitats [9]. This variety of different growth forms can be observed in the Pululahua Geobotanical Reserve, which has been identified as an area with an important diversity of plants, despite being a small, protected area (3356 ha) within the PANE (Heritage of Natural Areas of the State) [1]. This reserve has a plant nursery devoted exclusively to orchids that contains samples of most of the orchid species growing in the area.

The distribution and richness of orchids in Ecuador is restricted and is mainly due to human activities that disturb their habitat, especially in the Andean region, where recent studies show that deforestation puts the permanence of native ecosystems, such as montane forests, at risk due to the expansion of the agricultural and livestock frontier [10]. In addition, the specificity in the ecological relationships of orchids, especially with microorganisms and pollinators, in many cases limits their survival [11].

The abundance of orchids in natural habitats depends mainly on the compatibility of the interactions between the species, the physical environment in which they develop, and other organisms that share this environment [12,13]. For this reason, orchid species have evolved into different shapes, colors, and sizes that allow them to be classified as epiphytes, terrestrial, saprophytes, and lithophytes.

The main objective of the present study was to evaluate the similarity, abundance, and orchid diversity in two sectors of the Pululahua Geobotanical Reserve: the La Reventazón sector, which is located at 2019 m.a.s.l.; the Chaupisacha sector, which is located at 1963–2013 m.a.s.l. The results obtained made it possible to review protection priorities and conservation strategies, considering the biodiversity and threatened features of the area.

## 2. Materials and Methods

### 2.1. Study Area

The study was carried out in the Pululahua Geobotanical Reserve, which is located on the north-western side of Quito, the capital city of Ecuador (0°3′35.116″ N, −78°30′2.752″ W). It has a wide range of altitudes, between 1600 and 3356 m.a.s.l., since it is located on the side of the Pululahua volcano [14]. The main ecosystems in the area are the evergreen low mountain forest and the evergreen mountain forest, which correspond to the cloud forest ecosystem category [15]. The average annual rainfall ranges between 1000 and 1600 mm, while the mean temperature ranges between 12 and 19.5° C [16].

The Chaupisacha sector is a native area, part of the lower montane evergreen forest life zone, at coordinates 0°5′34.276″ N, −78°30′23.528″ W and at an altitude of 1963–2013 m.a.s.l. (Figure 1). It is characterized by the presence of 20–35 m forests canopy height, the majority of which are composed of trees with straight trunks, mainly from the Lauraceae, Rubiaceae, and Melastomataceae families [16]. They are found in the mountain relief in the sub-Andean part to the east of the Andes Mountain range, which has steeply inclined slopes.

The La Reventazón sector is located at coordinates 0°3′32.58″ N, −78°30′6.753″ W, at a height of 2195–2311 m.a.s.l. (Figure 1). It is characterized by hills with relatively steep slopes, from low to very high, rectilinear slopes, generally sharp peaks, and slopes greater than 40% by the Reventazón stream [16]. Some granitic and numerous intrusions, volcanic ash, undifferentiated metamorphic formations, and discontinuous pyroclastic cover have also been observed there. It is part of the evergreen montane forest life zone, which is characterized by having trees with gnarled branches and dense and compact crowns, and it is dominated by Andean elements, mainly from the Melastomataceae (Miconia), Solanaceae, Myrsinaceae, Aquifoliaceae, Araliaceae, and Rubiaceae. The sector was affected by landslides 100 years ago, whose forest is under natural recuperation. The site is surrounded on the northern, eastern, and southern borders by an area covered by gypsum, volcanic ashes, and quartz, mostly deprived of vegetation (part of the intervention area in Figure 1).

### 2.2. Data Collection and Analysis

The geolocalizations of the orchids in the field were introduced into the reserve’s database to update the register files of the Reserve Management Plan. Moreover, the existence of different forest types (evergreen low montane forest in Chaupisacha and Evergreen montane forest in La Reventazón), made it necessary to carry out quantitative and comparative analyses between plots. In these forests, three types of habits were registered for orchid species: epiphytes, terrestrial, and facultative epiphytes. The orchid density in each forest was estimated according to the number of individuals found and the area evaluated.

A botanical inventory of orchids was carried out in situ in two sectors: Chaupisacha and La Reventazón. Two plots of 1 ha were established, one in each sector (Figure 1). The method described in Gentry and Dodson (1987) [17] was followed as the first step in the field monitoring, carried out to determine the different indexes of abundance and orchid diversity. This sampling method was modified, setting a series of 10 contiguous 20 × 50 m transects instead of 2 × 50 m separated by 20 m among transects. Orchid individuals were identified by simple observation or by using binoculars. The use of ladders allowed access to the upper branches of host trees, and the use of poles with gathering forks in the extreme allowed the removal of orchids from host tree branches. Identification of species was made by consulting different orchids’ identification guides [2,6,8,15,18], the Tropicos data base of the Missouri Botanical Garden, and by comparison to samples from the Herbarium of Pontifical Catholic University of Ecuador (QCA).

Two individuals of each species were collected, with or without flowers. Those in the vegetative stage were maintained in a greenhouse until flowers emerged and their final identification could be made. After this, 1 sample will be deposited in the Herbarium of the Pontifical Catholic University of Ecuador; the second specimen will be returned to the plot where it was collected using the geographical coordinates that were registered on collection of the individuals.

### 2.3. Diversity

A quantitative analysis of the different plots made it possible to compare the orchid diversity of the different microhabitats in each sector and to compare them. The following parameters were calculated and compared. The first was Alpha diversity, which was calculated by applying the Simpson Dominance index (λ) [19]. This index calculates the relative abundance of the different species on the studied sites [20] and indicates the probability of two individuals taken randomly from the same species. Since its value is inverse to equity, the diversity can be calculated as 1 − λ [21].
λ = Σ p_i_^2^(1)

The second was the Shannon–Wiener diversity index (H′) [22], which was employed as a measure of the diversity that combines species richness (the number of species in a given area) with their relative abundances.
(2)$$H^{\prime }=-\sum_{i=1}^{N}\mathrm{pi}\;\ln\left(\mathrm{pi}\right)$$
where pi (=ni/N) is the relative abundance of species I, ni is the number of individuals of each species, and N is the total number of species.

The difference between the H’ index values obtained for the different localities was compared by estimating the variance and degrees of freedom, using the Student’s *t*-test, as described in [23].

In order to complement the Shannon–Wiener index, equity (J′) [24] was estimated as
J′ = H′/Hmax(3)
where Hmax = ln(N) measures the proportion of diversity in relation to the maximum expected in a community of N species. J′ varies between 0 and 1, such that 1 corresponds to situations in which all species are equally abundant.

### 2.4. Similarity and Dissimilarity

Beta diversity was used to compare the richness of plots and was estimated by employing the Jaccard Similarity Coefficient [25], which was calculated as follows:Ij = C/(A + B + C)(4)
where A and B are the number of species present solely in each respective community and C is the number of species present in both; Ij varies between 0 and 1.

Two other quantitative measures of similarity were employed to compare plots to support the results of this study: Euclidean distance [26,27]. The Euclidean distance of locations j and k (Δ_jk_) was calculated as:(5)$$\Delta_{\mathrm{jk}}=\surd \sum_{i=1}^{n}{\left(X_{\mathrm{ij}}-X_{\mathrm{ik}}\right)}^{2}$$
where:X_ij_ is the number of individuals of species i in location j,X_ik_ is the number of individuals of species i in location k,and n is the total number of species.

The Euclidean distance increases with the number of species in the samples, and the average distance was, therefore, calculated in order to compensate for this:(6)$$d_{\mathrm{jk}}=\surd \left(\Delta {\mathrm{jk}}^{2}\right)/n$$

The Euclidean distance varies from 0 to infinity: the larger the distance, the less similar the two plots are.

The Manhattan metric was standardized in [27] (by considering the distance between samples as the summed difference of the number of each shared species), signifying that it has a range from 0 (similar) to 1 (dissimilar).
(7)$$B=\frac{\sum_{i=1}^{n}\left(|X_{\mathrm{ij}}-X_{\mathrm{ik}}|\right)}{\sum_{i=1}^{n}\left(X_{\mathrm{ij}}+X_{\mathrm{ik}}\right)}$$
where B is the Bray–Curtis dissimilarity measure and X_ij_; X_ik_ and n are the same as in (5). The Bray–Curtis measure was then calculated as 1-B. The Bray–Curtis measure ignores cases in which the species is absent in samples obtained from both plots. It has the drawback that the results obtained are dominated by the most abundant species, signifying that species represented in low numbers add very little to the value of the coefficient.

### 2.5. Conservation of Orchids in Natural Settings Following the IUCN Scale

Finally, the analysis of the above parameters served to determine the conservation parameters of orchids in their natural settings, following the IUCN scale: EX = Extinct; EW = Extinct of nature; CR = Critically Endangered; EN = Endangered; VU = Vulnerable; NT = Near Threatened; LC = Least Concern; DD = Insufficient data NE = Not evaluated, [28]. The threat categories of each species were reviewed according to IUCN Categories and Criteria, which are displayed in the database of Ecuadorian and international herbaria, such as Herbarium of the Pontificial Catholic University of Ecuador and Tropicos from Missouri Botanical Garden. For each sample parameters were taken: population size reduction; geographic range size, fragmentation, location, decline andfluctuations; declining population size and subpopulations; restricted distribution and quantitative analysis of extinction risk [29].

## 3. Results

### 3.1. Diversity of the Forest in the Chaupisacha and La Reventazón Sectors

In Chaupisacha, 922 individuals were collected, which belonged to 55 species grouped into 24 genera. Most species were epiphyte (87.3%), six species had a terrestrial habit (10.9%), and only one species (1.8%) had a facultative epiphytic habit. Orchid density was low in this sector (0.96 ind./100 m^2^), as was Simpson dominance (λ) (0.097 ± 0.01), while Simpson’s diversity (1 − λ) attained 0.903 ± 0.01 and Shannon–Weaner Diversity (H′) attained 1.29 ± 0.03. Two species were observed in relatively large numbers: *Epidendrum embreei*, with 209 individuals, and *Xylobium leontoglossum*, with 149 individuals, while there were less than 80 individuals of the remaining species, which explains the relatively high equity in this sector (J′ 0.78) (Table 1).

In La Reventazón, a total of 9196 individual orchids were collected, which belonged to 46 species and 26 genera. Life habits were more balanced in this sector, with 13 epiphyte species (28.3%), 16 facultative epiphytic species (34.8%), and 17 terrestrial species (37%). Orchid density was high (183,92 individuals/100 m^2^), while Simpson dominance (λ) was low (0.15 ± 0.01 SD); Simpson’s Diversity attained 0.85 ± 0.01 and Shannon–Weaner Diversity (H′) attained 1.02 ± 0.005. In this sector, 12 species were represented from 158 to 2664 individuals, with an equity value of 0.61 (Table 1).

The difference between the Shannon–Weaner diversity indices, obtained for Chaupisacha (H′) 1.29 ± 0.03 and Reventazón (H′) 1.02 ± 0.005, was highly significant (Student *t*-test, *p* < 0.001), with Chaupisacha having a greater number of species and diversity, although the density of orchids in this sector was much lower than in the La Renventazón.

Only 14 species were found in both sectors, and each sector maintained its specificity, which was 41 species in Chaupisacha and 32 species in La Reventazón.

### 3.2. Similarity between the Forests in the Chaupisacha and La Reventazón Sectors

When considering the Jaccard index, the plots have little similarity, sharing only 14 species (16%) of the 87 observed in both communities (Table 2). The considerable Euclidean distance (1490), along with its average distance (398), revealed a high dissimilarity between the two sectors evaluated. The Bray–Curtis dissimilarity index confirmed very little similarity between the two evaluated sectors.

### 3.3. State of Conservation

According to IUCN [23], some of the species registered are presented in the conservation category as: (a) threatened to disappear from the wild: *Dracula felix* (Figure 2A), *Epidendrum diothonaeoidess* (Figure 2B); Least concern: *Phragmipedium lindenii* (Figure 2C), *Restrepia guttulata* (Figure 2D), and vulnerable to disappearing from the wild: *E. polyanthogastrium* (Figure 2E).

In Chaupisacha, five species with two types of threats were recorded. These were: Vulnerable (VU) and Near Threatened (NT): *Dracula felix* (Luer) Luer and *Masdevalliaan gulata* Rchb.f.; Critically Endangered (CR) and Vulnerable (VU) to disappear from the wild: *Epidendrum ornithoglossum* Schltr., Vulnerable (VU) *Scelochilus luerae* Dodson and with data deficient extinct (DD) (EX) *Stelis striolata* Lindl.

In La Reventazón, eight species were observed to be under threat, two species were recorded with two types of threats, i.e., Vulnerable (VU) and Near Threatened (NT) *D. felix*, *Byrsella angulata* (Rchb. f.) Luer; while there was one threat as Vulnerable (VU): *Epidendrum polyanthogastrium* Hágsater and Dodson; two as Near threatened (NT): *Epidendrum brachystele* Schltr., *Epidendrum polyanthogastrium* Hágsater and Dodson, and four as Least concern (LC): *Epidendrum diothonaeoides* Schltr., *Phragmipedium lindenii* (Lindl.) Dressler and N.H. Williams, *Pleurothallis macra* Lindl. and *Restrepia guttulata* Lindl.

Some factors that determine this condition could be the destruction of natural habitat, the growth of the agricultural frontier, and the excessive extraction of orchids from their natural setting.

Several species in the list were identified only at the genus level because equatorial orchids require further systematic studies for a correct identification or description of new species. They are maintained in greenhouses under cultivation in the reserve.

## 4. Discussion

This study evaluated the orchid composition of 1 ha plots and is the first study that involved such a large sample size. Results indicate a high richness of orchids in the Montane Forest of the Reventazón sector, with 9196 individuals and 46 species, and the Lower Montane Forest of the Chaupisacha sector, with 922 individuals and 55 species. Additionally, it shows that there is a marked difference between the individuals and species diversity in the two sectors. A comparison between the Chaupisacha sector, which has a very humid montane forest (bmh-M, according to [30]), an annual precipitation of 2132 mm, and an average temperature of 13–14 °C, and the Yanachaga Chemillen National Park, Pasco, Peru, is appropriate, since the latter has a predominance of steep rocky slopes covered by forests and is located between 460 and 3643 m.a.s.l. In the latter case, there are records of 470 individuals of the Orchidaceae family, distributed in 25 species and 14 genera [31]. It is evident that the Chaupisacha sector has double the number of individuals and species, while La Reventazón has 20 times more orchids than the protected area in Peru. Therefore, the limited distribution of orchid species in the La Reventazón and Chaupisacha sectors may be conditioned by the steep topography characteristic of the north of the Andes Mountains [32] and because they are within an area of volcanic activity.

Furthermore, a study in an Evergreen Montane Forest in the Cotapata National Park in Bolivia, which is located between 1500–2200 m.a.s.l. and characterized by steep slopes covered by vegetation, has reported a total of 38 species of orchids, 80% of them with an epiphytic habit [33]. The similarity between this forest and that of the Chaupisacha sector studied herein may, in terms of proportion of epiphytic species, be owed the location of both forests on steep slopes, which is a disadvantage for terrestrial species because of the instability of the forest substrate as a result of frequent landslides.

The evaluation of orchid habits in this study was represented from 28.3 to 87.3% epiphytes, 1.8 to 34.8% facultative epiphytes, and 10.9 to 37% terrestrial species. Previous studies carried out in Ecuador indicate that the number of epiphytic orchids in Ecuador exceeds that of the terrestrial group [2,34]. Epiphytic orchids are located mainly in the canopy of host trees, which is associated with little moss, but also on the fronds of terrestrial ferns and on dead or standing trunks in the forest. The predominance of the epiphytes was observed in Chaupisacha but not in La Reventazón. The forest in Chaupisacha is dominated by trees, while in La Reventazón, the vegetation is dominated by shrubs and trees of a low height. In a Tequendama very humid forest, located from 2000–2250 m.a.s.l. in central Colombia, a native and recuperated zone after logging, a similar study was performed comparing the orchid communities in two areas [35]. Results have shown that epiphytic habit predominated, with 81% of species, terrestrial 8%, lithophytes 7%, and facultative epiphytes or semi-terretrial 4%.

Environmental factors, such as humidity, precipitation, subtropical temperature, and abundant arboreal host ferns, favor the proliferation of terrestrial orchids in the La Reventazón sector, in contrast to the conditions of fewer trees with fissured bark and moss observed in Chaupisacha. Furthermore, the density was lower in Chaupisacha, but more orchid species were registered on the trees, possibly associated with the presence of fallen trees as substrate. In La Reventazón, the species richness was lower, but the density of orchids was much higher, with 122 individuals per 100 square meters.

Equity is associated with the way in which species are represented in numbers within a community [36]. The orchid species found in the forests were not uniformly represented (equity 0.78 and 0.61 in Chaupischa and La Reventazon, respectively). The environment with fallen trees could help any species to become dominant when compared to the primary forest environment with clearings. The clearings in the forest caused by fallen trees, possibly as a result of heavy storms, also contained more orchids than those covered with a higher density of trees. Despite the fact that the branches of the trees contained orchids, the amount of lighting in the environment probably affected the survival and reproductive capacity of the local orchids.

The distribution of orchids in the plots was heterogeneous in regards to the density and composition of the species, which were, according to the Jaccard index, reflected in the low similarity between Chaupisacha and La Reventazón. A more in-depth analysis of this case, using indices that employ the abundance of species in each sector, such as the Euclidean distance and the Bray-Curtis dissimilarity index, supported the low similarity of the sectors evaluated, in spite of the high number of species (87) involved in the analysis. The mean Euclidean distance was high (398), and the Bray–Curtis dissimilarity index was also high (0.906). These results show that both plots were evaluated in a montane forest and have particular characteristics regarding the distribution of epiphytes, facultative epiphytes, and terrestrial orchids within them. This can be associated with several environmental factors, such as being in different bands of height above sea level and the quality of the substrate [31]. A study in Montane forests in Cajamarca, Perú, between 800 and 2700 m.a.s.l., has shown that locations situated in closer height bands had lower Euclidean distances, in the order of 174, in contrast to the more separated locations, which had a Euclidean distance of 270 [37]. In the present study, the mean Euclidean distance between Chaupisacha and La Reventazón was 398, with a reduced difference in heights from 1963–2013 and 2200–2311 m.a.s.l., respectively. Such a large value of Euclidean distance between Chaupisacha and La Reventazón may be explained by factors other than distance, such as substrate quality and the characteristics of host trees.

Each plot was characterized by a unique set of orchid species, which suggests that orchid dispersion has limitations and that each group of individuals of a species within a sector behaves as a metapopulation, with a variable capacity to interact with other individuals within the forest [38]. Likewise, the comparison of orchid communities in native and recuperated areas in a Tequendama very humid forest, Colombia, found that richness (34 ad 38 species, respectively) and Simpson’s dominance (0.08 and 0.11) was similar in both areas, but Jaccards’s similarity was very low (0.17) [35].

The limited dispersion of orchids in the evaluated forests could be associated with several factors, such as: the availability of appropriate host tree species to contain orchids [39,40] (but see [41] for a non-significant role of orchid species distributed among tree species), the scarcity of pollinators, which could limit flower fertilization and seed production [42,43], or the lack of specific mycorrhizal symbiotic fungi that restricts orchid seed germination [44]. This phenomenon could be associated with the endemism observed in north-western Ecuador, where 20% of the reported species are endemic [32,45].

Changes in the density of orchids are associated with the type of forest and the presence of small trees and shrubs within it [12]. In the primary forest, orchids occupied the lower stratum (0.5–5 m) and middle stratum (5–10 m), and there was a rapid doubling of density and species richness towards the upper stratum (10-m). Only heliophytic species were found above 10 m. In the forest clearings in La Reventazón, most orchid species were found in the lower stratum (0–0.5 m), while in Chaupisacha, the lower stratum is less suitable for orchid survival owing to the presence of slippery and rocky soils, whose movement because of natural phenomena causes the loss of epiphytes.

Some of the species registered in this study have two of the IUCN [28] threat conservation categories, simultaneously [46], as is the case for *B. angulate* (NT, VU), *D. felix* (NT, VU), and *E. ornithoglossum* (CR, VU). CITES protects all orchid species by placing them in Appendix A and Appendix B. *Phragmipedium lindenii* is in Appendix A and it is thus in a higher conservation category.

Some factors that determine these threat conditions could be the destruction of the natural habitat by explosive natural factors, since the sectors studied are next to the Pululahua Volcano, which has geologically comprised volcanic and volcano-sedimentary complexes, with highly variable reliefs, in general, and very steep slopes of 30% to more than 70%, in addition to black soils of varying depths, resulting from volcanic ash, loamy, sandy loam, or sandy textures. On certain sites, there are rocky outcrops, with temperatures and humidity regimes in transition to per humid, in addition to anthropic factors, such as the excessive extraction of orchids from their natural settings [14].

Tree species that serve as hosts for epiphytic orchids are called phorophytes. Castillo-Pérez [46] mention that some orchids maintain a high degree of specificity or preference for their phorophytes; therefore, conservation efforts should also include these tree species.

## 5. Conclusions

Orchid richness in Chaupisacha included 55 species, of which 41 were exclusive species, while in La Reventazón, there were 46 species, of which 32 were exclusive. Orchid species were found to be highly dominant, and individual species formed isolated patches within the forest.

Orchids live on host tree branches and trunks up to 15 m high, but some trunks are dead, standing, or fallen. Umbrophilic orchid species occupied the lower part (0.5–4 m) of the primary forest, while Heliophytic species were found in both forests, occupying the middle (4–10 m) and high (10–15 m) strata.

Finally, conservation efforts and actions are a high priority for the evaluated forests in order to maintain their diversity and endemism and those of vascular plants in general. It is necessary to reduce the threat posed to wild forests by the growth of the agricultural frontier. Management plans are necessary for the conservation of species classified as threatened, and it should include the identification of natural populations, studies on their population dynamics in situ and ex situ, and conservation programs. These should be carried out with the help of the country’s botanical gardens, research institutions, and universities.

## Figures and Tables

**Figure 1 plants-11-00698-f001:**
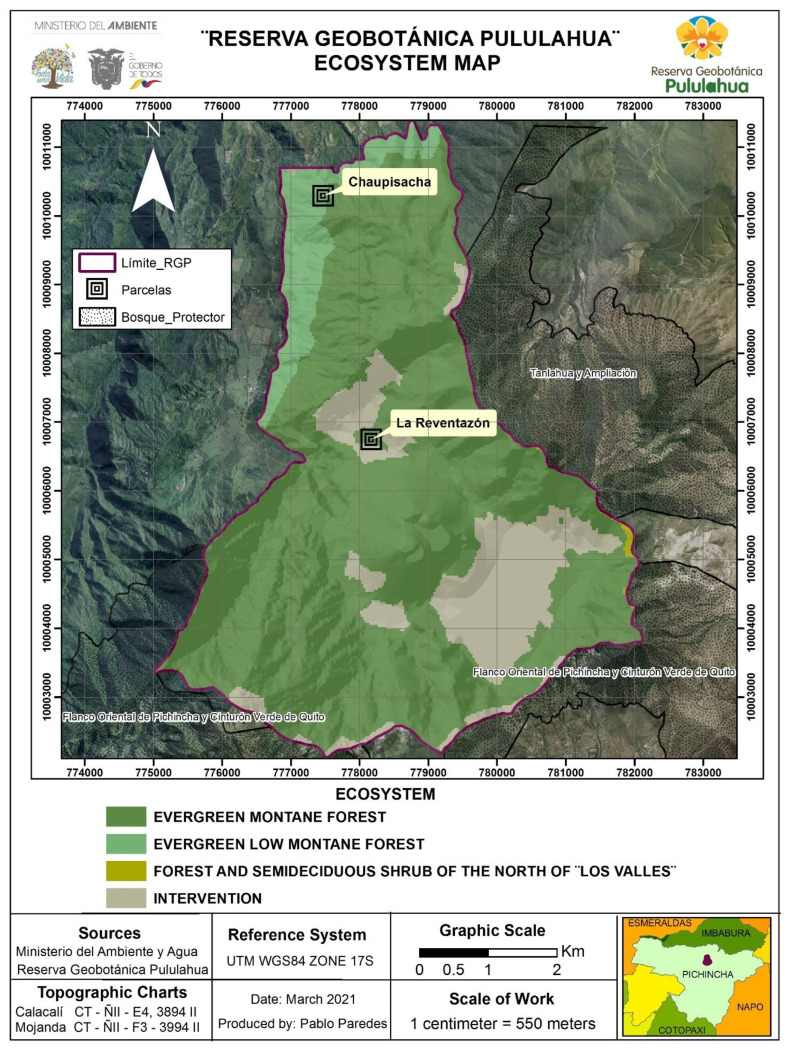
Map of the Pululahua Geobotanical Reserve showing the Chaupisacha and La Reventazon sectors. Source: [1,14].

**Figure 2 plants-11-00698-f002:**
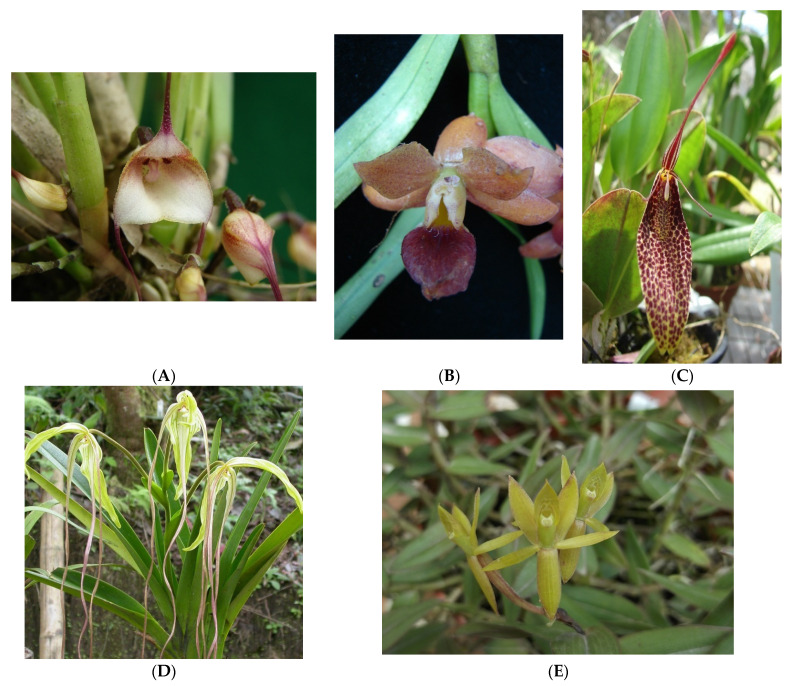
Threatened species in the Chaupisacha and La Reventazón sectors of Pululahua Geobotanical Reserve. (**A**). *Dracula felix*; (**B**). *Epidendrum diothonaeoidess*; (**C**). *Phragmipediumlindenii;* (**D**)*. Restrepia guttulate*; (**E**). *E. polyanthogastrium*.

**Table 1 plants-11-00698-t001:** Plot surface, richness, orchid density, and diversity indices in two forests on the Pululahua Geobotanical Reserve, Pichincha province, Ecuador.

Parameter	Chaupisacha	La Reventazón
Plot surface (m^2^)	10,000	10,000
No. orchid individuals	922	9196
No. exclusive species	41	32
No. shared species	14	
Density (ind./100 m^2^)	0.96	122
Simpson’s diversity (1 − λ)	0.90	0.85
Shannon-Weaner (H′) *	1.29 ^a^	1.02 ^b^
Equity (Pielou’s J′)	0.78	0.61

* Student’s *t*-test. Different letters imply significant differences (*p* < 0.001).

**Table 2 plants-11-00698-t002:** Similarity indices of the forests in Chaupisacha and La Reventazón, Pululahua Geobotanical Reserve, Pichincha province, Ecuador.

Index	Index Value
Jaccard	0.161
Euclidean distance	1490
Mean Euclidean distance	398
Bray-Curtis (dissimilarity)	0.906

## Data Availability

Data are shown in the Appendix A and Appendix B included with this manuscript.

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
