# Peer review of "Analysis of the Orchidaceae Diversity in the Pululahua Reserve, Ecuador: Opportunities and Constraints as Regards the Biodiversity Conservation of the Cloud Mountain Forest"

_plants, 2022, doi:10.3390/plants11050698_

Round 1

Reviewer 1 Report

Line 61: It has a wide range of altitudes, between

Line 71:  forests canopy height

Line 88: … terrestrial, and facultative epiphytes.

Line 95: …in each sector and to compare them.

Line 97: … on the studied sites…

Line 97-98: … taken randomly from the same species.

Line 148: …while about the remaining species, ….

Line 154: 12 species were represented from 158 to 2664 individuals….

Line 168: … between the two evaluated sectors.

Line 208: … studied herein may, in terms of proportion of epiphytic species, be owed to

Line 215: predominance of the epiphytes

Line 219: … was lower in Chaupisacha, but more orchid…

Line 275: … for the evaluated forests ….

Line 275: , and it should include ….

Author Response

Line 61: It has a wide range of altitudes, between

Corrected. Line 70. “It has a wide range of altitudes, between 1600 and 3356 m.a.s.l, since it is located on the side of the Pululahua volcano”.

Line 71:  forests canopy height

Corrected. Line 77. “It is characterized by the presence of 20-35 m forests canopy height, the majority of which are composed of trees with straight trunks, mainly from the Lauraceae, Rubiaceae and Melastomataceae families”.

Line 88: … terrestrial, and facultative epiphytes.

Corrected. Line 95. “In these forests, three types of habits were registered for orchid species: epiphytes, terrestrial, and facultative epiphytes”.

Line 95: …in each sector and to compare them.

Corrected. Line 115-116. “A quantitative analysis of the different plots made it possible to compare the orchid diversity of the different microhabitats in each sector and to compare them.”

Line 97: … on the studied sites…

Corrected. Line 117-118. “This index calculates the relative abundance of the different species on the studied sites”.

Line 97-98: … taken randomly from the same species.

Corrected. Line 118-119. “This index calculates the relative abundance of the different species on the studied sites [20] and indicates the probability of two individuals taken randomly from the same species”.

Line 148: …while about the remaining species, ….

Corrected. Line 172. Two species were observed in relatively large numbers: Epidendrum embreei, with 209 individuals, and Xylobium leontoglossum, with 149 individuals, while about the remaining species, there were less than 80 individuals, which explains the relatively high equity in this sector (J’ 0.78) (Table 2; Annex 1).

Line 154: 12 species were represented from 158 to 2664 individuals….

Corrected. Line 179. In this sector, 12 species were represented from 158 to 2664 individuals, with an equity value of 0.61 (Table 1; Annex 2).

Line 168: … between the two evaluated sectors.

Corrected. Line 194.

Line 208: … studied herein may, in terms of proportion of epiphytic species, be owed to

Corrected. Line 237-239. The similarity between this forest, and that of the Chaupisacha sector studied herein may, in terms of proportion of epiphytic species, be owed the location of both forests on steep slopes, which is a disadvantage for terrestrial species because of the instability of the forest substrate as a result of frequent landslides.

Line 215: predominance of the epiphytes

Corrected. Line 245.

Line 219: … was lower in Chaupisacha, but more orchid…

Corrected. Line 245.

Line 275: … for the evaluated forests ….

Corrected. Line 315.

Line 275: , and it should include ….

Corrected. Line 315.

Reviewer 2 Report

The article is well written and interesting and makes original contributions on orchid diversity and conservation issues.

However, I give some suggestions to improve it

First of all in the title why do you write Orquidaceae ? It is more correct as in the rest of the text to write Orchidaceae or possibly orchids.

Line 69-70, 74-74 here the text is burdened by the indication of three different geographic coordinates of the two study areas. I suggest to put only one geographic coordinate, let's say the central part , for each study site and rather than the UTM coordinates the geographic coordinates in degrees; this facilitates the quick search of the site, for example on Google Earth

Line 84 you should specify which texts you used for the taxonomic identification of the studied species

Line 136-137 Conservation of orchids in natural settings following the IUCN scale

you say that you used diversity parameters to assign IUCN risk categories. but to ascertain the risk categories it is necessary to have, and I believe you do, other information such as population size and possible declines, size of the AOC fragmentation, etc. as specified in IUCN Standards and Petitions Committee. See  Guidelines for Using the IUCN Red List Categories and Criteria. Version 15. 2022 Prepared by the Standards and Petitions Committee. Downloadable from https://www.iucnredlist.org/documents/RedListGuidelines.pdf. Therefore you need to better specify the methodology used to assign categories

line 173 3.3 State of conservation

here too, the motion of the risk category should be better understood, for example decline of population reduction of the area of presence in the  region or simply use of an already existing red list for Ecuador.

Appendix A and  B

in these two lists about  the IUCN status column it is not correct to indicate the inclusion in the CITES appendix, this is another thing concerning the legal protection of these species that could be mentioned in the text; so if there is no IUCN category just put a hyphen

I also note that many species in the list are determined only at the genus level. this is an important shortcoming but it can be justified by the fact that equatorial orchids require further systematic studies for a correct determination or description of new species. This too could be commented in the text to explain to the reader that it is not a negligence  but rather the possible initiation of further research.

With the suggested improvements the manuscript will be worthy of being published on Plants

Author Response

REFEREE 2 COMMENTS

The article is well written and interesting and makes original contributions on orchid diversity and conservation issues.

However, I give some suggestions to improve it

First of all in the title why do you write Orquidaceae ? It is more correct as in the rest of the text to write Orchidaceae or possibly orchids.

Done. Line 2.

Line 69-70, 74-74 here the text is burdened by the indication of three different geographic coordinates of the two study areas. I suggest to put only one geographic coordinate, let's say the central part, for each study site and rather than the UTM coordinates the geographic coordinates in degrees; this facilitates the quick search of the site, for example on Google Earth

Done.Lines 70; 76 and 80.

Line 84 you should specify which texts you used for the taxonomic identification of the studied species

Done. Specification of botanical guides and references of them were included in the text. Line 104 and in the reference list.

Line 136-137 Conservation of orchids in natural settings following the IUCN scale. You say that you used diversity parameters to assign IUCN risk categories. but to ascertain the risk categories it is necessary to have, and I believe you do, other information such as population size and possible declines, size of the AOC fragmentation, etc. as specified in IUCN Standards and Petitions Committee. See Guidelines for Using the IUCN Red List Categories and Criteria. Version 15. 2022 Prepared by the Standards and Petitions Committee. Downloadable from https://www.iucnredlist.org/documents/RedListGuidelines.pdf. Therefore you need to better specify the methodology used to assign categories.

Done. Explanation was included in the text. Lines 158-164.

line 173 3.3 State of conservation

Here too, the motion of the risk category should be better understood, for example decline of population reduction of the area of presence in the region or simply use of an already existing red list for Ecuador.

Done. Now it is better explained in the State of conservation section, specially in Lines 164-166.

Appendix A and B

in these two lists about  the IUCN status column it is not correct to indicate the inclusion in the CITES appendix, this is another thing concerning the legal protection of these species that could be mentioned in the text; so if there is no IUCN category just put a hyphen

I also note that many species in the list are determined only at the genus level. this is an important shortcoming but it can be justified by the fact that equatorial orchids require further systematic studies for a correct determination or description of new species. This too could be commented in the text to explain to the reader that it is not a negligence but rather the possible initiation of further research.

Done. CITES quotations were eliminated and hyphens were added in both appendices. A comment on the large number of species identified only to genera was included in the text. Lines 220-222.

With the suggested improvements the manuscript will be worthy of being published on Plants

Reviewer 3 Report

This is an interesting and valuable case study providing novel information about diversity of species of Orchidaceae family in specific mountain forests. Authors sampled two forest sites in Ecuador and calculated various diversity indices. The results are documented (providing species lists and diversity values). However, more details would be necessary about the field sampling. I also suggest improving the study background and discussion and providing more information about similar studies.

Introduction was well written about the general research background. However, I miss information about previous inventories (in Ecuador and in other countries) which used the same methodology. Authors cited only 3 papers (24, 25 and 27) from Peru and Bolivia. It was not clear whether these studies sampled also 1 ha with exactly the same methods. Any difference in sampling parameters (plot size, plot shape, method of searching orchid individuals, the number of times the study site was visited for sampling) could influence results and could make studies not-comparable. Sampling 1 ha areas fully for all orchid individuals was hard work that should be appreciated explicitly. It would be important to know if this very detailed study was the first in Ecuador and how many similar datasets exist globally.

Authors refer Gentry and Dodson (Biotropica 1987, 19: 149-156.) for sampling details. I checked this paper and it was not clear how Gentry and Dodson method could be used in the present study. The cited study (Gentry and Dodson) sampled 0.1 ha - but not 1 ha. The cited study sampled all individuals of all plant species - not only orchids. The cited study sampled several 2 x 50 m transects - but it is not clear how the 1 ha plots were sampled in this study. Please, clarify how Gentry and Dodson method was used here and what was the shape of the 1 ha plots (100 m x 100 m rectangular area or transects)? How authors could prove that really all orchid individuals were found in these very complex forests? How the 1 ha areas were selected (random or preferential)? According to Figure 1, one plot was close to an intervention area. Why did you choose this area?

All orchids were sampled. Appendix A and B shows the “Frequency” of species. However, the meaning “frequency” could be else than total number of individual per species – I guess these numbers refer the number of individuals found. Please, be clear about it (or explain what subplots were used for calculating frequencies).

Diversity studies often found many rare species with only 1 individual (singletons). In case of rare species, how authors could collect two individuals from each sector? (cf. L 83.) How many individuals were in vegetative stages? Please, clarify if all vegetative individuals found in the forests were collected (how many) and grown in greenhouse.

Methods

Diversity indices were properly described: However, please correct the terminology: use “orchid diversity” instead of “diversity” in general, and communicate beta diversity as dissimilarity “between plots” or “between samples” instead of “between communities”. In case you refer specific “orchid communities”, the related concepts should be introduced and discussed.

In this study density of orchid species and diversity of orchid collections were estimated. Please, explain how these data were used to estimate conservation states (describing how particular species were threatened or vulnerable). It is not enough citing a reference (see my concerns about the inaccurate citation of Gentry and Dodson 1987 paper). Any case, readers should be able to understand the methods by reading this particular paper.

Discussion

I like the explanations for diversity patterns and for differences between sites. However, only two sites were compared. In this special case, all differences between sites could be interpreted as “causes”. Consequently, these ideas remain speculative without citing more studies with similar findings.

I was not convinced that dispersal limitation was a main factor explaining between-site differences (L 244). “Spatial dispersion” is not synonyms to “spatial dispersal”. I think the spatial dispersion of species were really limited (but not the dispersal). In fact, Authors provided many (other) potential causes that could explain the high beta diversity found.

Small comments

Title: “Orquidaceae” should probably be “Orchidaceae”

Keywords: delete “dispersal”

Figure 1. Please improve the quality. It was hard to read.

L 122-123: “(or biomass)” Was biomass of orchids sampled?

Table 2 Bray-Curtis similarity + Bray-Curtis dissimilarity = 1 (you can delete one of them)

L 237: The cited paper: Calatayud, G. Diversity of Orchidaceae family in the montane forest in San Ignacio (Cajamarca, Peru). Rev. Per. Biol. 2005, 12(2), 309-316. has a single authors, so “authors” should be “author”.

L 270: “Orchid richness in Chaupisacha included 55 more species than in Reventazón”This means that in Chaupisacha you would have 46 + 55 species. Please, correct.

Author Response

REFEREE 3 COMMENTS

This is an interesting and valuable case study providing novel information about diversity of species of Orchidaceae family in specific mountain forests. Authors sampled two forest sites in Ecuador and calculated various diversity indices. The results are documented (providing species lists and diversity values). However, more details would be necessary about the field sampling. I also suggest improving the study background and discussion and providing more information about similar studies.

Thanks for the appreciation and for the valuable comment. Now we give more explanations about in the Introduction section. Lines 54-66. “The distribution and richness of orchids in Ecuador is restricted and is mainly due to human activities that disturb their habitat. Especially in the Andean region where recent studies show that deforestation puts the permanence of native ecosystems such as montane forests at risk due to the expansion of the agricultural and livestock frontier (Jadán et al., 2016). In addition, the specificity in the ecological relationships of orchids, especially with microorganisms and pollinators, in many cases limits their survival (Endara et al., 2010).

The abundance of orchids in natural habitats depends mainly on the compatibility of the interactions between the species, the physical environment in which they develop and other organisms that share this environment (Newman et al., 2007; Whigham and Willems, 2003). For this reason, orchid species have evolved in different shapes, colors and life sizes that allow them to be classified as epiphytes, terrestrial, saprophytes and lithophytes”.

Introduction was well written about the general research background. However, I miss information about previous inventories (in Ecuador and in other countries) which used the same methodology. Authors cited only 3 papers (24, 25 and 27) from Peru and Bolivia. It was not clear whether these studies sampled also 1 ha with exactly the same methods. Any difference in sampling parameters (plot size, plot shape, method of searching orchid individuals, the number of times the study site was visited for sampling) could influence results and could make studies not-comparable. Sampling 1 ha areas fully for all orchid individuals was hard work that should be appreciated explicitly. It would be important to know if this very detailed study was the first in Ecuador and how many similar datasets exist globally.

This study is the first in Ecuador with this methodology to determine the richness and abundance of orchids. Epiphytes have generally been studied establishing transects of 20 x 50m, (0.1 ha in each), according to the method used by Gentry and Dodson (1987) to determine the diversity of epiphytic plants. The comparison was made with the studies from Bolivia and Peru because they maintained the same methodology in the 1 ha sampling site and evaluated only the epiphytes. Now a better discussion is made in Lines 270-285. Also, the importance of the present study is underlined in the Discussion section Lines 224-225.

Authors refer Gentry and Dodson (Biotropica 1987, 19: 149-156.) for sampling details. I checked this paper and it was not clear how Gentry and Dodson method could be used in the present study. The cited study (Gentry and Dodson) sampled 0.1 ha - but not 1 ha. The cited study sampled all individuals of all plant species - not only orchids. The cited study sampled several 2 x 50 m transects - but it is not clear how the 1 ha plots were sampled in this study. Please, clarify how Gentry and Dodson method was used here and what was the shape of the 1 ha plots (100 m x 100 m rectangular area or transects)? How authors could prove that really all orchid individuals were found in these very complex forests? How the 1 ha areas were selected (random or preferential)? According to Figure 1, one plot was close to an intervention area. Why did you choose this area?

The plot selected in La Reventazón was affected by landslides 100 years ago whose forest is under natural recuperation. The site is surrounded on the northern, eastern and southern borders by an area covered by gypsum, volcanic ashes and quartz, mostly deprived of vegetation (part of the Intervention area in Fig. 1). Now in the M&M section the methodology is better explained. Lines 98-101.

All orchids were sampled. Appendix A and B shows the “Frequency” of species. However, the meaning “frequency” could be else than total number of individual per species – I guess these numbers refer the number of individuals found. Please, be clear about it (or explain what subplots were used for calculating frequencies)

The frequency is the number of individuals of each species found in the plot. All orchids were counted, with the following consideration, plants with caespitose pseudobulbs were considered 1 individual for every 3 pseudobulbs. Explanations in the text are in Lines 106-110.

Diversity studies often found many rare species with only 1 individual (singletons). In case of rare species, how authors could collect two individuals from each sector? (cf. L 83.) How many individuals were in vegetative stages? Please, clarify if all vegetative individuals found in the forests were collected (how many) and grown in greenhouse.

Only 2 individuals of each species were collected, with or without flowers. Those in vegetative stage were maintained in a greenhouse until flowers emerge and their final identification could be made. After this, 1 sample will be deposited in the Herbarium of the Pontifical Catholic University of Ecuador; the second specimen will be returned to the plot where it was collected using the geographical coordinates that were registered on collection of the individuals. Fifteen species were found with only 1 specimen in both study sites. Of these, 10 were identified to species and the remaining 5 species were taken to the greenhouse. Explanations in the text are in Lines 106-110.

Methods

Diversity indices were properly described: However, please correct the terminology: use “orchid diversity” instead of “diversity” in general and communicate beta diversity as dissimilarity “between plots” or “between samples” instead of “between communities”. In case you refer specific “orchid communities”, the related concepts should be introduced and discussed.

Done through all the manuscript. Lines 41, 63, 94, 99 and 115.

In this study density of orchid species and diversity of orchid collections were estimated. Please, explain how these data were used to estimate conservation states (describing how particular species were threatened or vulnerable). It is not enough citing a reference (see my concerns about the inaccurate citation of Gentry and Dodson 1987 paper). Any case, readers should be able to understand the methods by reading this

The conservation stage of orchid species was determined from data bases in herbaria and literature information. The information on species density in the plots was not used for this purpose. Explanations about are in Lines 154-166.

Discussion

I like the explanations for diversity patterns and for differences between sites. However, only two sites were compared. In this special case, all differences between sites could be interpreted as “causes”. Consequently, these ideas remain speculative without citing more studies with similar findings.

Now more sites are compares and more references are cited to support the explanations provided through all the discussion section. Lines 270-308.

I was not convinced that dispersal limitation was a main factor explaining between-site differences (L 244). “Spatial dispersion” is not synonyms to “spatial dispersal”. I think the spatial dispersion of species were really limited (but not the dispersal). In fact, Authors provided many (other) potential causes that could explain the high beta diversity found.

Corrected We agree that between-site differences in species composition could be due to several factors, dispersal among them, but also substrate changes, amount of fallen tree trunks, clears in the forests, microclimate provided by host tree and quality of bark, the presence of other epiphytes, among others. Discussion is made at Lines 279-289.

Title: “Orquidaceae” should probably be “Orchidaceae” Done. Line 2.

Keywords: delete “dispersal” Done.

Figure 1. Please improve the quality. It was hard to read. Done. A new figure with a better quality is provided.

L 122-123: “(or biomass)” Was biomass of orchids sampled? Text removed

Table 2 Bray-Curtis similarity + Bray-Curtis dissimilarity = 1 (you can delete one of them) Done

L 237: The cited paper: Calatayud, G. Diversity of Orchidaceae family in the montane forest in San Ignacio (Cajamarca, Peru). Rev. Per. Biol. 2005, 12(2), 309-316. has a single authors, so “authors” should be “author”. Done

L 270: “Orchid richness in Chaupisacha included 55 more species than in Reventazón”This means that in Chaupisacha you would have 46 + 55 species. Please, correct. The statement in Conclusions was corrected. Lines 310-312.

Round 2

Reviewer 3 Report

I care fully checked the revision. The authors reponded my comments and made a satisfying revision. I think, the paper can be accepted.

I suggest few small amendments that can be done during proof reading.

Authors use the word „recuperated” for succession and restoration that should be changed to the commonly used terms.

L 94 „natural recuperation” should be „natural recovery”

L 373 „recuperated zone” should be „restored zone”

L 428 „recuperated area” should be „restored area”